# Case-control study of autonomic symptoms in the setting of Long COVID with tilt table testing

Matthew S. Durstenfeld[1,2,3*], Nirosh Mataraarachchi[4], Michael J. Peluso[1,5], Marta Levkova-Clark[1,2], Veronica Schaffer[3], Emily A. Fehrman[1,5], Grace Anderson[1,5], Diana Flores[1,5], Timothy J. Henrich[1,6], Carlin S. Long[1,7], Steven G. Deeks[1,4], Priscilla Y. Hsue[3]

1 Department of Medicine, University of California, San Francisco, California, United States of America, 2 Division of Cardiology, Zuckerberg San Francisco General, San Francisco, California, United States of America, 3 Division of Cardiology, University of California, Los Angeles, United States of America, 4 School of Medicine, University of California, San Francisco, United States of America, 5 Division of HIV, Infectious Diseases, and Global Medicine, Zuckerberg San Francisco General Hospital, University of California, San Francisco, United States of America, 6 Division of Experimental Medicine, Zuckerberg San Francisco General Hospital, University of California, San Francisco, United States of America, 7 Division of Cardiology, UCSF Health, University of California, San Francisco, United States of America

* matthew.durstenfeld@ucsf.edu

## Abstract

### Background

Autonomic symptoms and orthostatic syndromes have been reported in Long COVID, but few studies have characterized findings using head up tilt table testing.

### Objective

To characterize autonomic responses to positional changes among individuals with Long COVID.

### Methods

We assessed autonomic symptoms using the Composite Autonomic Symptom Scale 31 (COMPASS 31) instrument and performed head up tilt table testing for 30 minutes at 70 degrees among individuals with Long COVID and recovered comparators.

### Results

We included 26 participants (median age 56 years, 50% female median 25 months after first COVID): 16 with Long COVID and 10 recovered comparators. COMPASS 31 scores (0–100, higher is worse) were higher among those with Long COVID (median 30.5 vs 8, p = 0.003). Heart rate was 8 beats per minutes higher throughout tilt among those with Long COVID (95% CI 1.1 to 14.4; p = 0.02); there were no differences in blood pressure. Ten (63%) with Long COVID had symptoms during tilt compared to none among recovered participants (p = 0.003). Three (19%) with Long

**Data availability statement:** Deidentified study data are available and stored at the Open Science Foundation at: https://osf.io/wyxcz/?view_only=f53e436ba04a424a9653865938f42b36.

**Funding:** This study was primarily funded by a philanthropic gift from Charles W. Swanson to CL. MSD was funded by NIH/NHLBI K12 HL143961 and K23HL172699. PYH was supported by NIH/NAID 2K24AI112393. The funders played no role in the study design, data collection and analysis, decision to publish, or preparation of the manuscript. Its contents are solely the responsibility of the authors and do not necessarily represent the official views of the NIH.

**Competing interests:** M.S.D: consulting fees from Merck. P.Y.H.: modest honoraria from Gilead and Merck and research grant from Novartis unrelated to the submitted work. M.J.P.: consulting fees from Gilead Sciences, AstraZeneca, BioVie, and Apellis Pharmaceuticals, and research support from Aerium Therapeutics, outside the submitted work. S.G.D.: consulting for Enanta Pharmaceuticals and Pfizer and research support from Aerium Therapeutics outside the submitted work. T.J.H.: consulting fees for Roche and Regeneron outside the submitted work. All other authors report no relevant disclosures or conflicts. This does not alter our adherence to PLOS ONE policies on sharing data and materials.

COVID had clinically abnormal findings: one each with orthostatic hypotension, and delayed orthostatic hypotension, and cardioinhibitory/vasovagal presyncope.

## Conclusions

Among those with chronic autonomic symptoms in the setting of Long COVID, symptoms were common during tilt testing, and heart rate was increased, but most did not meet diagnostic criteria for a clinically abnormal hemodynamic response. Further research into mechanisms of autonomic symptoms in Long COVID is urgently needed.

## Introduction

Autonomic disorders including postural orthostatic tachycardia syndrome (POTS), orthostatic hypotension (OH), and orthostatic intolerance without tachycardia or hypotension have been reported following COVID-19 as post-acute sequelae of COVID-19 or symptomatic Long COVID [1,2]. The role of autonomic dysfunction as a contributor to Long COVID symptomatology is uncertain, but there is overlap between common Long COVID symptoms and common autonomic symptoms. A survey of individuals recruited through Long COVID support groups reported that two thirds of survey participants had a Composite Autonomic Symptom 31 (COMPASS-31) score >20, suggestive of autonomic dysfunction [3]. More severe autonomic symptoms are associated with worse quality of life and higher levels of disability among those with Long COVID [4]. Orthostatic intolerance is included in the diagnostic criteria for myalgic encephalitis/chronic fatigue syndrome (ME/CFS), which overlaps with Long COVID especially in its most disabling forms [5].

Head up tilt-table testing (HUTT) is the gold-standard objective test for assessing orthostatic symptoms in the setting of autonomic dysfunction, but existing studies have major limitations. A case-series of 24 individuals with cardiovascular phenotype post-acute sequelae of COVID-19, or Long COVID, referred from a Long COVID clinic, reported that 23/24 had "orthostatic intolerance" during HUTT. In that study, 15 (65%) had normal tests prior to administration of nitroglycerin and 22% were on beta blockers [6]. Beta blockers lower sensitivity and nitroglycerin lowers specificity for clinical symptoms [7]. Importantly, all 15 with nitroglycerin-provoked orthostatic intolerance reported that their Long COVID symptoms were improving or resolved, suggesting that pharmacologic provocation may not be the most clinically useful approach. Both beta blockers and nitroglycerin modify physiologic responses to orthostatic stressors, so the relevance of this study's findings on the mechanisms of symptoms in Long COVID is challenging to interpret.

Other studies in different populations and using different methodologies have reported that 0–30% of individuals met criteria for POTS [8–11]. Those studies which reported higher prevalences were conducted in referral centers for autonomic dysfunction among individuals referred for clinical concern for Long COVID and/or specifically for autonomic dysfunction [10,11]. Several studies used the active stand test

or shortened 10-minute head up tilt testing [8,9,11]. In the context of the considerable variability of the findings reported, we conducted a case-control study of individuals with Long COVID, with and without postural symptoms consistent with autonomic dysfunction, compared to individuals who had recovered from COVID.

## Methods

### Setting, participants, and study design

As previously reported, the Long-term Impact of Infection with Novel Coronavirus (LIINC) study is a COVID recovery cohort of adults that includes longitudinal symptom assessment [12]; it includes participants with and without Long COVID recruited from April 2020 through the present. We have previously reported findings using echocardiography, cardiac magnetic resonance imaging, cardiopulmonary exercise testing, and ambulatory cardiac rhythm monitoring [13,14]. Here we report findings from a case-control sub-study within LIINC focused on autonomic function using COMPASS 31 and HUTT as the primary assessments. Participants for this case-control study were recruited from June 14, 2022, through May 9, 2024.

For our case-control study, we included participants without cardiovascular or autonomic disease diagnosed prior to COVID who were diagnosed with COVID-19 at least 6 months prior confirmed with nucleic acid amplification or lateral-flow antigen-based testing. For the "case" group, we specifically recruited individuals who had participated in the LIINC cardiovascular sub-study who reported at least one cardiovascular Long COVID symptom including chest pain, dyspnea, palpitations, fatigue, or decreased exercise capacity and who answered "Yes" to a survey question at least 3 months after COVID: "Do you have any symptoms when you change positions?" From the same cohort, we recruited a sample of adults with at least one cardiovascular Long COVID symptom including chest pain, dyspnea, palpitations, fatigue, and decreased exercise capacity who answered "No" to the survey question. Finally, we recruited a sample of individuals recovered from COVID-19 who reported that they had fully recovered from COVID without any persistent symptoms as a comparator group, matched by age within five years and sex to the "case" group. We excluded two individuals taking medications that modify the heart rate response (beta blockers, nondihydropyridine calcium channel blockers, and ivabradine); we did not exclude individuals for use of other antihypertensive medications or other medications that may impact autonomic function. No included participants had participated in a formal exercise rehabilitation program prior to HUTT or had been treated for POTS.

### Assessment of autonomic symptoms

We used the validated 31-question Composite Autonomic Symptom Score (COMPASS 31) as our primary instrument to assess chronic autonomic symptoms [15]. It is scored from 0 to 100, with a higher score representative of a higher burden of autonomic symptoms encompassing orthostatic intolerance, vasomotor, secretomotor, gastrointestinal, bladder, and pupillomotor domains. The COMPASS 31 instrument and its scoring have been validated against the 169-item Autonomic Symptom Profile and its validated 84-question instrument, COMPASS [15]. While COMPASS-31 was developed to assess autonomic failure, it has been used to characterize autonomic function in POTS and other conditions, including Long COVID [3]. We also specifically considered the orthostatic intolerance domain, which has a maximum weighted score of 40 points.

### Head up tilt table testing

We conducted unmedicated head up tilt table testing (HUTT). Participants were instructed to fast and abstain from caffeine and nicotine. Participants changed into a hospital gown and were prepared with electrocardiogram leads, defibrillator pads, an antecubital peripheral intravenous line, safety straps, and covered with a blanket. Participants then rested quietly in a supine position for 5–10 minutes on the tilt table. Then a resting supine 12 lead electrocardiogram was recorded. Noninvasive intermittent blood pressure with an automated blood pressure cuff over the brachial artery was recorded at

least 3 times at rest. Then the table was tilted to 70 degrees for 30 minutes. Heart rhythm and pulse oximetry was monitored continuously. Blood pressure was recorded every 2 minutes. After the 30-minute measures were obtained, the participant was returned to supine and monitored for at least 5 minutes or until symptom resolution and return to baseline vitals for up to 20 minutes. The upright portion was terminated by returning the participant to supine prior to 30 minutes for syncope, pre-syncope with severe symptoms, or concerning heart rhythm changes defined as pauses >5 seconds or ventricular tachycardia, or participant request. Intravenous fluids could be administered if symptoms did not resolve, and hemodynamics were not restored with return to supine at the discretion of the supervising physician. We did not provoke syncope with the use of medications. HUTT was performed and symptoms were recorded by trained nursing staff under supervision of cardiac electrophysiologists.

We planned to report the number and percentage who met diagnostic criteria for adult POTS in accordance with the consensus clinical definition which requires a 30 beat per minute increase in heart rate within 10 minutes of tilt in the absence of orthostatic hypotension, in the presence of chronic orthostatic symptoms lasting 3 months or longer [16–18]. We defined orthostatic hypotension as a reduction in systolic blood pressure of at least 20 mm Hg or diastolic blood pressure of at least 10 mm Hg occurring within 3 minutes of tilt with or without symptoms [19], and delayed orthostatic hypotension if it occurred more than 10 minutes after tilt. We also reported individuals who had a greater than 30 beat per minute increase in heart rate and those whose heart rate went above 120 beats per minute in the absence of orthostatic hypotension or syncope. We used the average of the supine measures, the 10-minute measure, the 30-minute measure (immediately before return to supine), and the first return to supine measure as our primary measures of interest, but we reviewed all measurements and plotted the supine average, 2-minute, 4-minute, 6-minute, 8-minute, 10-minute, 20-minute, 30-minute, and first return to supine measures. All studies were reviewed immediately by the supervising electrophysiologists and then subsequently by the first author (MSD) to apply the research case definitions.

### Other correlative data

As previously reported, we have conducted echocardiography, cardiopulmonary exercise testing, and ambulatory heart rhythm monitoring in subsets of LIINC participants, including some who participated in this sub-study [13,14].

### Statistical analysis

We report descriptive statistics (number, proportion) by Long COVID status with exploratory analyses further stratified by the answer to our survey question about symptoms with position change. Given its skewed distribution, we compared the COMPASS 31 score and Orthostatic Domain score by Long COVID status using the Wilcoxon rank sum test and report the median and interquartile ranges. We compared categorial variables (symptoms during tilt, abnormal tilt, for example) using the Fisher exact test. We used logistic regression models with symptoms during HUTT and clinically abnormal HUTT as outcomes and COMPASS 31 score and Orthostatic Domain scores included as predictors in separate models. We used linear mixed effects models to model change in heart rate and blood pressure over time using random effects per participant, interaction terms for time and Long COVID status, and robust standard errors. REDCap was used for COMPASS 31 and participant questionnaires. Statistical analyses were performed in STATA version 17.0 and additional visualization was conducted using R version 4.2.0 using the ggplot2 library.

### Ethical approval and data availability

The University of California San Francisco Committee on Human Research approved of this study (#20–33000). All participants provided written informed consent. Deidentified study data are available and stored at the Open Science Foundation at: https://osf.io/wyxcz/?view_only=f53e436ba04a424a9653865938f42b36.

## Results

We enrolled 26 participants (median age 56 years, 50% female, Table 1) who completed HUTT at median 25 months after initial SARS-CoV-2 infection (range 9−38 months) which occurred between March 2020 and July 2022. All but one participant were vaccinated prior to the study visit; 5/26 (19%) received their first vaccination prior to their first known SARS-CoV-2 infection. We recruited 16 participants who reported ongoing Long COVID symptoms including 9 who answered "Yes" to our survey question about experiencing symptoms with position change. We also recruited 10 recovered comparators without Long COVID selected to be age- (within 5 years) and sex-matched with the group who answered "Yes" to our survey question about symptoms with position change.

### Autonomic symptom burden assessed with COMPASS 31

Median COMPASS 31 total score (0–100, higher is worse) was 30.5 (interquartile range 20–42, total range 1–65) among those with Long COVID, compared to 8 among recovered comparators (interquartile range 5–17, total range 0–22; p = 0.003, Fig 1). Among those with any cardiovascular Long COVID symptoms, COMPASS 31 total scores did not vary based on responses to our survey question regarding symptoms with position change (S1 Fig; p = 0.83).

The median orthostatic domain sub-score out of 40 possible points was 18 (interquartile range 12–28, total range 0–28) among those with Long COVID compared to 0 among recovered participants (interquartile range 0–0, total range 0–12; p = 0.004). Orthostatic domain score did not vary based on responses to our survey question regarding symptoms with position change (S1 Fig; p = 0.49).

### HUTT findings

Among participants with Long COVID, 10 (63%) experienced symptoms during HUTT compared to none among recovered participants (p = 0.003). Symptoms included dizziness/lightheadedness (9), nausea (6), diaphoresis/sweating (3), headache (2), chest pain and numbness/tingling (2 each). One participant reported that HUTT triggered an episode of post-exertional malaise. Each 1-point increase in the COMPASS 31 orthostatic domain sub-score was associated with 11% higher odds of symptoms during tilt (OR 1.11, 95% CI 1.01–1.23, p = 0.04)

At baseline, the average supine heart rate was 8 beats per minute higher among participants with Long COVID, (95% Confidence Interval (CI) −1–16, p = 0.07, Fig 2). On average, heart rate increased by 7 beats per minute at 10 minutes (p < 0.001 vs baseline), by 11 beats per minute at 30 minutes (p < 0.001 vs baseline) and returned to baseline after

**Table 1. Participant characteristics.**

| | Long COVID (n = 16) | Recovered (n = 10) | P value |
|---|---|---|---|
| Age, years[a] | 54.4 ± 9.4 | 57.9 ± 11.3 | 0.68[d] |
| Female Sex[b] | 7 (44%) | 6 (60%) | 0.49[e] |
| Body Mass Index[a] | 29.6 ± 8.5 | 25.3 ± 4.0 | 0.20[d] |
| Months since First COVID[a] | 23 ± 6 | 28 ± 6 | 0.15[d] |
| Hospitalized for Acute COVID[b] | 3 (19%) | 1 (10%) | 1.00[e] |
| COMPASS 31 Total Score[c] | 30.5 (20, 42) | 8 (5, 17) | 0.005[f] |
| Orthostatic Domain Score[c] | 18 (12, 28) | 0 (0, 0) | 0.004[f] |
| HR baseline[a] | 66 ± 12 | 58 ± 5 | 0.02[g] |
| SBP baseline[a] | 128 ± 14 | 124 ± 15 | 0.39[g] |
| DBP baseline[a] | 76 ± 9 | 75 ± 8 | 0.88[g] |

Footnotes: [a]mean ± SD, [b]number (percentage), [c]median (interquartile range), [d]Two-sample T test, [e]Fischer's Exact, [f]Wilcoxon Rank Sum, [g]Mixed Effects model.

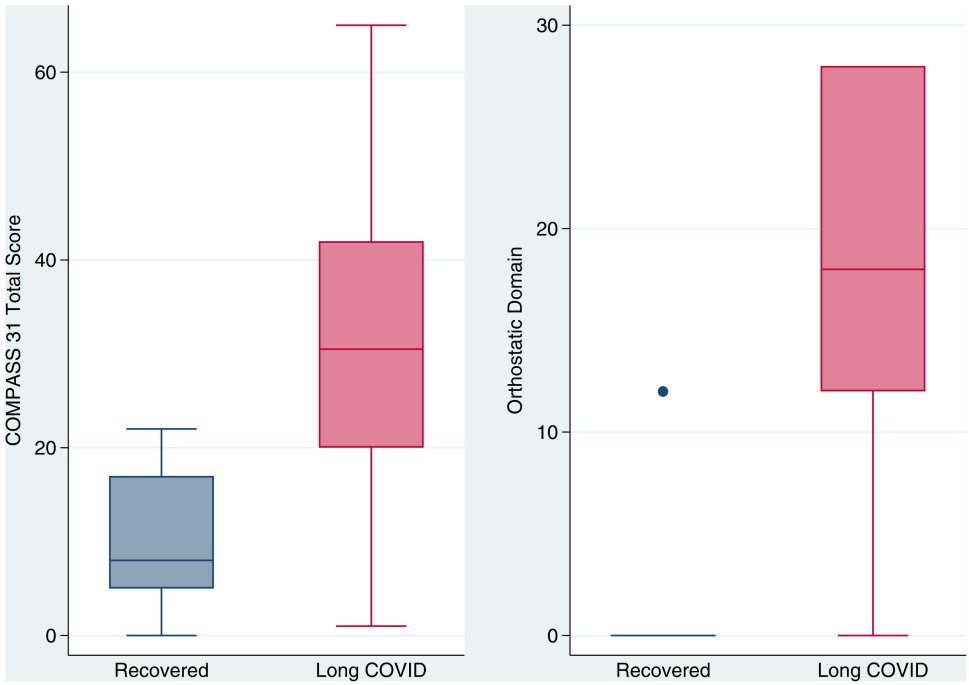

**Fig 1. COMPASS 31 total score and orthostatic domain scores by long COVID.** Boxplots of COMPASS 31 Total Score (left) and Orthostatic Domain Scores (right) by Long COVID versus Recovered.

returning to supine (p = 0.50 vs baseline). The average heart rate was 8 beats per minute higher among people with Long COVID across all time points (95% CI 1.1 to 14.4; p = 0.02). There was no significant difference in the change in heart rate over time by Long COVID status ($p_{interaction}$ = 0.34 overall; p > 0.4 for each individual timepoint). Similarly, at baseline the systolic blood pressure was 6 mm Hg higher among those with Long COVID, which was not statistically significant (p = 0.30), and systolic blood pressure neither significantly changed at 10 or 30 minutes compared to baseline, nor did change vary by Long COVID status (p = 0.84). Diastolic blood pressure was similar at all time points with no difference in change with tilt by Long COVID status (p = 0.86). Results were similar in analyses stratified by analysis of the survey response to presence of orthostatic symptoms (S2 Fig).

Three individuals (all with Long COVID symptoms) had clinically-evident abnormal blood pressure responses during HUTT (Fig 3). One had orthostatic hypotension (Baseline BP 130/72 HR 53 to BP 103/68, HR 81 with dizziness at 2 minutes), one had delayed orthostatic hypotension (Baseline BP 130/73 HR 56 to BP 86/56 HR 59 with nausea at 26 minutes), and one had delayed cardioinhibitory/vasovagal presyncope with a drop in blood pressure to 84/49 and heart rate to 45 at 25 minutes. No participants in our study met diagnostic criteria for POTS. No participants had a heart rate over 120 or even a transient increase in heart rate greater than 30 beats per minute.

### Exploratory: Cardiopulmonary exercise testing, ambulatory rhythm monitoring, and echocardiograms

Among those who had previously completed CPET (n = 19: 14 with Long COVID and 5 Recovered), the peak oxygen consumption on maximal effort upright cycle ergometry CPET was 38% lower on the percent predicted scale (95% CI 16–62) among those with Long COVID compared to recovered individuals (p = 0.003). There were trends toward lower adjusted heart rate reserve (64 vs 88%, p = 0.06), a measure of chronotropic response, and heart rate recovery at 1 minute (9 vs 16 beats per minute, p = 0.17) among those with Long COVID. Interestingly, both participants who experienced orthostatic

## Heart Rate

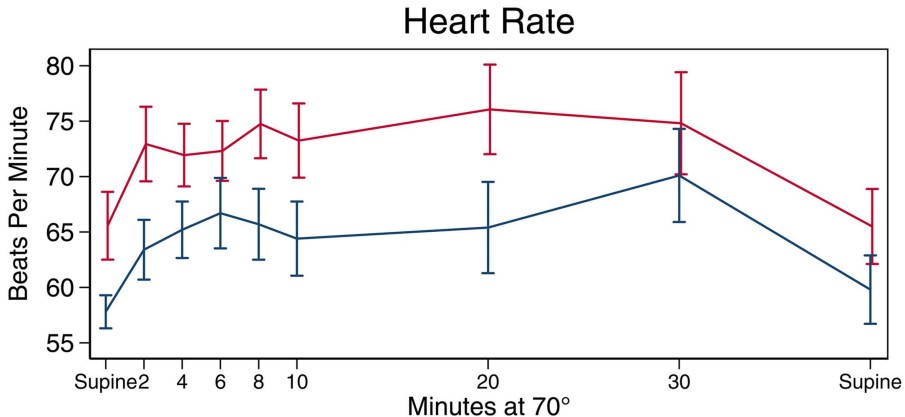

## Systolic Blood Pressure

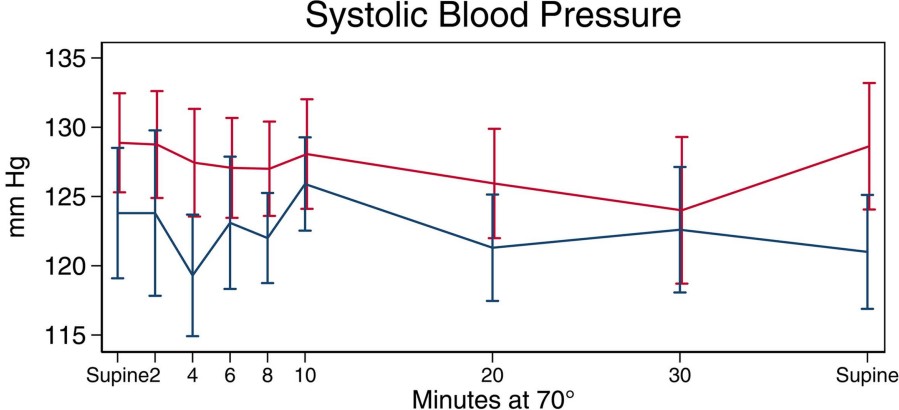

## Diastolic Blood Pressure

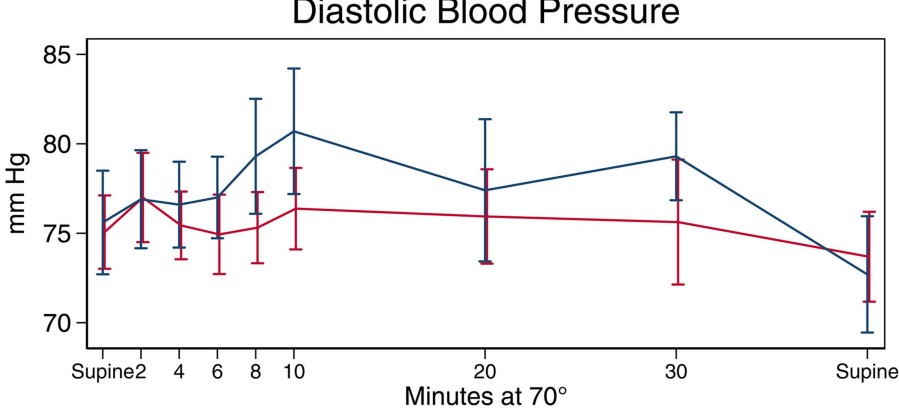

**Fig 2. Average heart rate and blood pressure during HUTT by Long COVID status.** Heart rate (top), systolic blood pressure (middle), and diastolic blood pressure (bottom) by Long COVID (red) versus Recovered (blue). Values are group means with error bars plus/minus standard error. Heart rate was elevated at baseline and remained elevated throughout HUTT but with no difference in change in heart rate by group, whereas blood pressure did not vary by group.

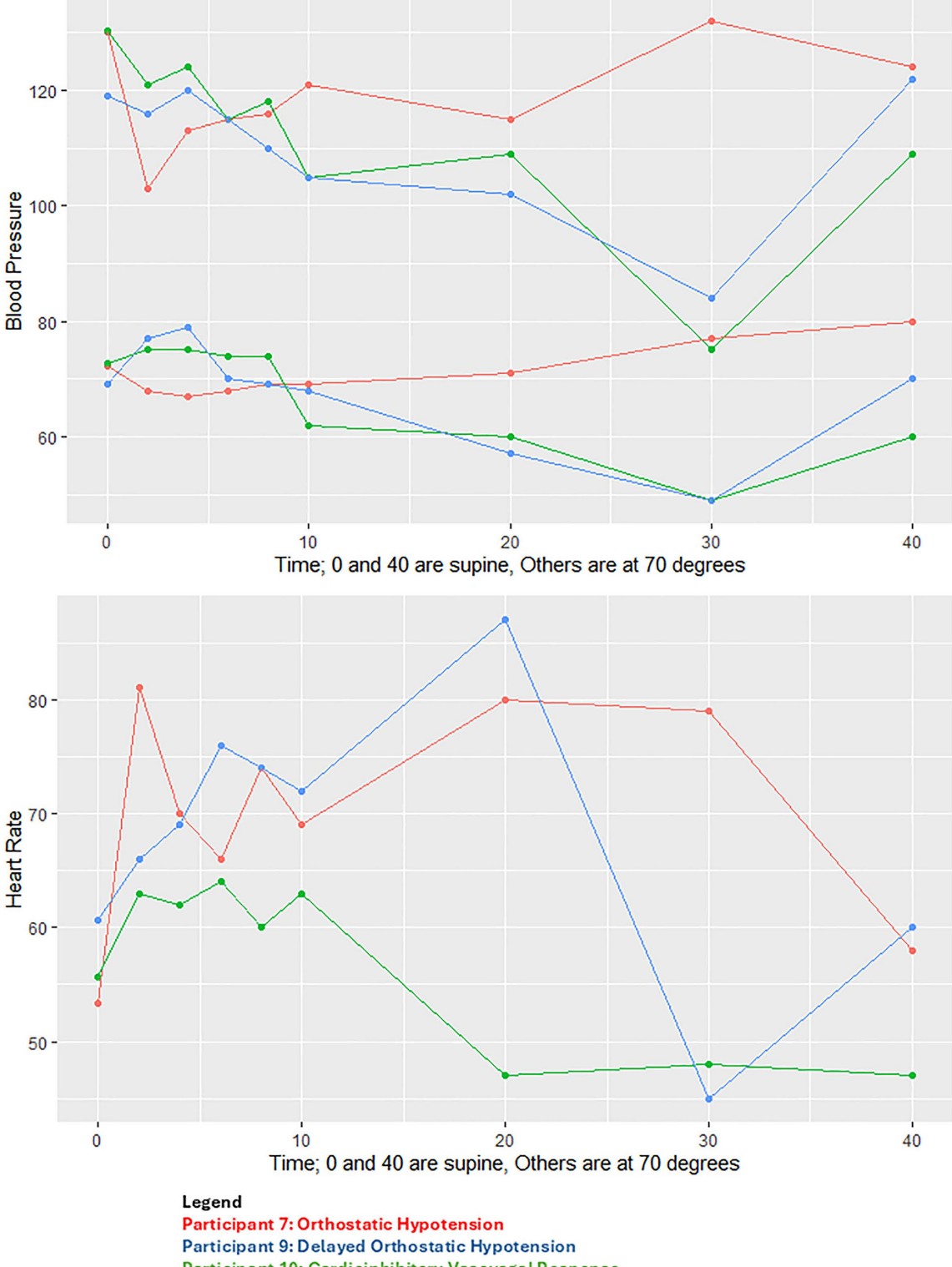

**Fig 3. Individual participant level heart rate and blood pressure for the three participants with abnormal hemodynamic responses.** Fig 3 Legend—Participant Level Heart Rate and Blood Pressure During Head Up Tilt Table Testing among the Three Participants with Abnormal Hemodynamics: Red = Orthostatic Hypotension, Blue = Delayed Orthostatic Hypotension; Green = Cardioinhibitory Vasovagal Response.

hypotension and delayed orthostatic hypotension had chronotropic incompetence on CPET, but the individual with a vaso-vagal/cardioinhibitory response had a normal cardiopulmonary exercise test.

Among 12 participants with ambulatory heart rhythm monitoring data (8 with Long COVID and 4 Recovered), there was a trend toward lower heart rate variability among those with Long COVID (standard deviation n-to-n 126 vs 160, p = 0.11). Twenty-four participants had previous research echocardiograms (16 with Long COVID and 7 Recovered). There were no differences in mean left ventricular ejection fraction (p = 0.48), left ventricular end-diastolic volume index (p = 0.21), stroke volume (p = 0.83), left ventricular mass index (p = 0.15), global longitudinal strain (p = 0.84), cardiac index (p = 0.90), or myocardial work (p = 0.91).

## Discussion

In our case-control study comparing individuals with Long COVID with individuals who had recovered from COVID without cardiopulmonary symptoms, we found higher COMPASS 31 scores among those with Long COVID. Chronic autonomic symptoms were common and nearly two thirds with Long COVID experienced symptoms during HUTT. We also found that people with Long COVID had higher average heart rates, but most individuals, even those with symptoms, did not meet clinical consensus definitions for POTS or orthostatic hypotension.

Like several other small studies [9,10,20], we found that orthostatic symptoms were common during HUTT even without meeting clinical criteria for an abnormal heart rate or blood pressure response during HUTT. Our findings that orthostatic symptoms are common are also consistent with several studies that used the NASA lean test in the setting of Long COVID [21–23].Our findings that some participants experience orthostatic hypotension or delayed orthostatic hypotension are consistent with several prior reports [9,11,24]. The delayed findings we observed suggest that a 10-minute HUTT, as recommended in the consensus guidance statement on the assessment and treatment of autonomic dysfunction in patients with post-acute sequelae of SARS-CoV-2 [26], may be insufficiently short to identify abnormalities in some individuals, although the increase in sensitivity from a longer test may decrease specificity [27]. Our findings also suggest that pharmacologic provocation is unnecessary to induce symptoms.

POTS certainly occurs post-COVID, as we see in our clinical practices. We were surprised that we did not identify anyone who met the consensus definition for POTS, which is consistent with several other reports with notable methodologic concerns but contrasts sharply from others. Of the two prior studies that reported low prevalences of POTS, one used the active stand test [8], which despite being included in the consensus definition is known to have different test characteristics for POTS [27,28]. The other did not use the consensus definition of POTS and returned participants to supine at the onset of symptoms but before an abnormal heart rate response had time to occur [9]. Our study did not directly compare the active stand test or the NASA lean test with the gold-standard HUTT. There is a major physiologic difference in active standing, which requires activation of skeletal muscles, compared to passive HUTT, and there is controversy over whether the NASA lean test is adequate in this clinical setting [25]. Our study, which used HUTT and the consensus definition of POTS, adds more rigorous evidence that many individuals with Long COVID with chronic autonomic symptoms do not meet the consensus definition of POTS.

Two studies with rigorous assessment of autonomic function with HUTT conducted at highly-respected autonomic centers found higher prevalences of POTS. A Canadian study led by the Calgary Autonomic Investigation and Management Clinic found that 30% of their study participants with post-acute sequelae of COVID-19 meet criteria for POTS [11]. Similarly, a retrospective analysis of patients at the Mayo Clinic found that 22% of patients met diagnostic criteria for POTS [10]. By recruiting at referral centers, the prevalence of POTS observed in those two studies is likely to be overestimated due to selection bias. Our case-control design and recruitment strategy does not allow estimation of prevalence estimates of Long COVID-POTS within the community, so this remains an outstanding research need.

Our correlative data raise questions regarding potential contributing mechanisms to Long COVID symptoms. Heart rate was consistently higher among people with Long COVID, which may be a sign of altered resting sympathetic/

parasympathetic tone. We did not find evidence of decreased left ventricular size, mass index, or cardiac output suggestive of cardiac deconditioning. There was a trend toward reduced heart rate variability among those with Long COVID; heart rate variability is a marker of autonomic health. This is concerning because reduced heart rate variability is associated with mortality and cardiovascular morbidity [29]. Most interestingly, both participants with orthostatic hypotension had evidence of chronotropic incompetence on CPET, and it seems plausible orthostatic hypotension may overlap with those with "preload failure" on CPET [30,31]. One prior study found that individuals with Long COVID have evidence of small fiber neuropathy and reduced cerebral blood flow during HUTT [32]. Contrasting with that study, another small study of autonomic function among individuals with Long COVID POTS compared to healthy controls found reduced cardiovagal modulation but normal sympathetic cardiac and vasoconstrictive functions [33]. Interestingly, a study using a sheep model found that impaired vagal function may result in reduced exercise capacity, reduced cardiac blood flow, and chronotropic incompetence [34].

There are several ongoing clinical trials investigating vagal nerve stimulation for Long COVID (clinicaltrials.gov NCT05421208, NCT05630040, NCT05918965) including two with a sham stimulation comparison group [35]. In POTS, transcutaneous vagal nerve stimulation improves postural tachycardia and lowers inflammatory cytokines compared to a sham control [36]. These studies may inform our understanding not only of Long COVID, but also other post-infectious autonomic syndromes.

### Limitations

Our primary limitations arise from our small sample size and case-control design (which limits our ability to comment on prevalence). We did not measure continuous beat-to-beat blood pressure either noninvasively or with invasive hemodynamic monitoring, or cerebral blood flow, which reduced our sensitivity for detecting hemodynamic changes, nor did we assess comprehensive autonomic function using the quantitative sudomotor axon reflex test, skin biopsy, or assess heart rate responses to Valsalva maneuver or deep breathing to assess the contributions of the parasympathetic or sympathetic axes of the autonomic nervous system. Correlative data such as echocardiography, cardiopulmonary exercise testing, and ambulatory heart rhythm monitoring were not available for all participants.

### Conclusion

Our COMPASS 31 findings are consistent with prior reports that autonomic symptoms are common in Long COVID, and we found higher average heart rates among people with Long COVID during HUTT. In the setting of Long COVID, many individuals who experience autonomic symptoms including orthostatic intolerance may appear to have "normal" heart rate and blood pressure responses during clinical head-up tilt table testing, especially if continuous blood pressure and cerebral blood flow are not measured— this should not be misinterpreted to mean that the symptoms are not real. Further research into mechanisms and treatment of autonomic dysfunction in the setting of Long COVID and the effect of potential therapies on autonomic symptoms is urgently needed.

### Supporting information

**S1 Fig. COMPASS 31 scores and orthostatic domain scores by screening questionnaire for symptoms with position change.** Boxplots of COMPASS 31 Total Score and Orthostatic Domain Scores by Recovered (blue, left), Long COVID with "No" answer to question about symptoms with position change (lavendar, middle), and Long COVID with "Yes" answer to question about symptoms with position change (orange, right).
(EPS)

**S2 Fig. Heart rate and blood pressure during HUTT by Long COVID group.** Group mean of heart rate and blood pressure during HUTT by Recovered (blue), Long COVID with "No" answer to question about symptoms with position

change (lavendar), and Long COVID with "Yes" answer to question about symptoms with position change (orange). Error bars are standard error of the mean.
(EPS)

**S3 Fig. Graphical abstract.** Among those with Long COVID compared to recovered comparator participants, we found higher COMPASS 31 scores, higher resting heart rate, more symptoms during head up tilt table testing, and that only some individuals meet diagnostic criteria for an abnormal hemodynamic response during tilt.
(TIF)

## Acknowledgments

We would like to acknowledge the LIINC Study participants and the LIINC study team for assisting with participant recruitment. We would also like to acknowledge the support of the UCSF Health Division of Electrophysiology and the UCSF Health Division of Cardiology ECG/Stress Testing Laboratory for conducting the tilt table testing.

## Author contributions

**Conceptualization:** Matthew S. Durstenfeld, Michael J. Peluso, Timothy J. Henrich, Carlin S. Long, Steven G. Deeks, Priscilla Y. Hsue.

**Data curation:** Matthew S. Durstenfeld.

**Formal analysis:** Matthew S. Durstenfeld.

**Funding acquisition:** Matthew S. Durstenfeld, Carlin S. Long, Priscilla Y. Hsue.

**Investigation:** Matthew S. Durstenfeld, Nirosh Mataraarachchi, Michael J. Peluso, Marta Levkova-Clark, Veronica Schaffer, Emily A. Fehrman, Grace Anderson, Diana Flores.

**Methodology:** Matthew S. Durstenfeld, Michael J. Peluso, Timothy J. Henrich, Steven G. Deeks, Priscilla Y. Hsue.

**Project administration:** Matthew S. Durstenfeld, Marta Levkova-Clark.

**Resources:** Matthew S. Durstenfeld, Michael J. Peluso, Carlin S. Long, Steven G. Deeks, Priscilla Y. Hsue.

**Software:** Matthew S. Durstenfeld.

**Supervision:** Steven G. Deeks, Priscilla Y. Hsue.

**Visualization:** Matthew S. Durstenfeld.

**Writing – original draft:** Matthew S. Durstenfeld.

**Writing – review & editing:** Nirosh Mataraarachchi, Michael J. Peluso, Marta Levkova-Clark, Veronica Schaffer, Emily A. Fehrman, Grace Anderson, Diana Flores, Timothy J. Henrich, Carlin S. Long, Steven G. Deeks, Priscilla Y. Hsue.

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
