## [Decision Letter · Decision Letter 0]

3 Sep 2025

Dear Dr. Durstenfeld,

Thank you for submitting your manuscript to PLOS ONE. After careful consideration, we feel that it has merit but does not fully meet PLOS ONE’s publication criteria as it currently stands. Therefore, we invite you to submit a revised version of the manuscript that addresses the points raised during the review process.

**ACADEMIC EDITOR: Justify the study background**

We look forward to receiving your revised manuscript.

Kind regards,

Jeyasakthy Saniasiaya, MD, MMed ORLHNS, FEBORLHNS

Academic Editor

PLOS ONE

Journal Requirements:

“This study was primarily funded by a philanthropic gift from Charles W. Swanson to CL. MSD was funded by NIH/NHLBI K12 HL143961 and K23HL172699 (MSD). PYH was supported by NIH/NAID 2K24AI112393. The funders played no role in the study design, data collection and analysis, decision to publish, or preparation of the manuscript. Its contents are solely the responsibility of the authors and do not necessarily represent the official views of the NIH.” 

“M.S.D: consulting fees from Merck. P.Y.H.: modest honoraria from Gilead and Merck and research grant from  Novartis unrelated to the submitted work. M.J.P.: consulting fees from Gilead Sciences, AstraZeneca, BioVie, and Apellis Pharmaceuticals, and research support from Aerium Therapeutics, outside the submitted work. S.G.D.: consulting for Enanta Pharmaceuticals and Pfizer and research support from Aerium Therapeutics outside the submitted work. T.J.H.: consulting fees for Roche and Regeneron outside the submitted work. All other authors report no relevant disclosures or conflicts.”

We note that one or more of the authors are employed by a commercial company: Merck, Novartis, Gilead Sciences, AstraZeneca, BioVie, and Apellis Pharmaceuticals, Enanta Pharmaceuticals and Pfizer

“This study was primarily funded by a philanthropic gift from Charles W. Swanson to CL. MSD was funded by NIH/NHLBI K12 HL143961 and K23HL172699 (MSD). PYH was supported by NIH/NAID 2K24AI112393. The funders played no role in the study design, data collection and analysis, decision to publish, or preparation of the manuscript. Its contents are solely the responsibility of the authors and do not necessarily represent the official views of the NIH. **“**

“This study was primarily funded by a philanthropic gift from Charles W. Swanson to CL. MSD was funded by NIH/NHLBI K12 HL143961 and K23HL172699 (MSD). PYH was supported by NIH/NAID 2K24AI112393. The funders played no role in the study design, data collection and analysis, decision to publish, or preparation of the manuscript. Its contents are solely the responsibility of the authors and do not necessarily represent the official views of the NIH.”

5. Please include captions for your Supporting Information files at the end of your manuscript, and update any in-text citations to match accordingly. Please see our Supporting Information guidelines for more information: http://journals.plos.org/plosone/s/supporting-information .

Additional Editor Comments (if provided):

It is definitely an interesting article which will add value to the current research on COVID-19 studies. However, Please justify the background of the study and how tilt table test adds value.

Reviewers' comments:

Reviewer's Responses to Questions

**Comments to the Author**

1. Is the manuscript technically sound, and do the data support the conclusions?

Reviewer #1: Partly

Reviewer #2: Yes

2. Has the statistical analysis been performed appropriately and rigorously?

Reviewer #1: Yes

Reviewer #2: Yes

3. Have the authors made all data underlying the findings in their manuscript fully available?

Reviewer #1: Yes

Reviewer #2: Yes

4. Is the manuscript presented in an intelligible fashion and written in standard English?

Reviewer #1: Yes

Reviewer #2: Yes

Reviewer #1: Hello.

This is a good article and the topic chosen is important and special and has been studied in a proper way in this article.

I thank the authors.

I hope we will read more and better quality studies from them in the future.

Good luck.

Reviewer #2: The study titled ‘Case-control study of autonomic symptoms in the setting of long COVID with tilt table testing’ by Durstenfeld et al. presents several intriguing findings. To further strengthen the manuscript, I recommend the following revisions:

1. Page 3, Introduction Section: The introduction outlines prior findings and notes inconsistencies in the literature; however, it does not clearly highlight why this represents a significant clinical or scientific problem worth investigating. Beyond mentioning variability in prevalence estimates, the rationale for the study is not compelling. The authors should better articulate the importance of clarifying autonomic dysfunction in Long COVID, to more convincingly justify the need for the present study.

2. Page 3, Lines 51-55: When discussing the prior case series, the manuscript briefly notes the influence of beta blocker use and nitroglycerin provocation on HUTT outcomes. However, this limitation is not sufficiently emphasized. The authors should explicitly state that these methodological issues undermine the validity of the reported prevalence of orthostatic intolerance and contribute to inconsistencies across previous studies. This clarification is important to strengthen the rationale for why the present case-control study is warranted.

3. Page 3, Lines 58-60: When comparing prior studies, the statement that “other studies in different populations and using different methodologies have reported 0–30% meeting criteria for POTS” is too general. For a meaningful comparison, the authors should at least specify what these populations were and briefly note the study contexts, so that readers can better understand the reasons for variability and how the present study fits within the existing literature.

4. Pages 5-6, Lines 101-130: The HUTT protocol does not state whether participants were fasting. Please clarify, as recent food or caffeine intake could affect autonomic responses.

5. Pages 5-6, Head up tilt table testing: The HUTT includes several observer-dependent measurements during HUTT. It is unclear whether these were performed by a single investigator or multiple individuals. The authors should clarify this.

6. Table 1: Some variables are reported as mean, while others are reported as median. This distinction is not noted in the footnotes. For clarity, the footnotes should specify which variables are summarized by mean ± SD and which by median (IQR).

7. Page 13, Lines 255-266: While the authors highlight some methodological concerns in prior studies reporting POTS post-COVID, the discussion would benefit from a more detailed explanation of these issues. For example, the limitations of the active stand test and the implications of returning participants to supine before a full heart rate response occurs should be elaborated to help readers better understand why those studies may have underestimated or misclassified POTS.

**Do you want your identity to be public for this peer review?** For information about this choice, including consent withdrawal, please see our Privacy Policy

Reviewer #1: No

Reviewer #2: No

---

## [Author Response · Author response to Decision Letter 1]

18 Sep 2025

Response to Reviewers

ACADEMIC EDITOR: Justify the study background

Authors’ Response: We have added additional information to the introduction to justify the study background.

Journal Requirements:

Authors’ Response: We corrected the formatting and file naming. We did not find instructions for including a graphical abstract. Nonetheless, we have included a graphical abstract which can be removed at the editors’ discretion in case it can be included.

“This study was primarily funded by a philanthropic gift from Charles W. Swanson to CL. MSD was funded by NIH/NHLBI K12 HL143961 and K23HL172699 (MSD). PYH was supported by NIH/NAID 2K24AI112393. The funders played no role in the study design, data collection and analysis, decision to publish, or preparation of the manuscript. Its contents are solely the responsibility of the authors and do not necessarily represent the official views of the NIH.”

Authors’ Response: We have included an amended statement which clarifies that additional funding was from internal division funds, and we now include the statement, “There was no additional external funding received for this study.”

“M.S.D: consulting fees from Merck. P.Y.H.: modest honoraria from Gilead and Merck and research grant from Novartis unrelated to the submitted work. M.J.P.: consulting fees from Gilead Sciences, AstraZeneca, BioVie, and Apellis Pharmaceuticals, and research support from Aerium Therapeutics, outside the submitted work. S.G.D.: consulting for Enanta Pharmaceuticals and Pfizer and research support from Aerium Therapeutics outside the submitted work. T.J.H.: consulting fees for Roche and Regeneron outside the submitted work. All other authors report no relevant disclosures or conflicts.”

We note that one or more of the authors are employed by a commercial company: Merck, Novartis, Gilead Sciences, AstraZeneca, BioVie, and Apellis Pharmaceuticals, Enanta Pharmaceuticals and Pfizer.

Authors’ Response: We would like to clarify that none of the authors are employed by a commercial company, although several of the authors have had limited consulting engagements with commercial entities outside of the submitted work, which we have been careful to disclose. We have included a slightly revised version in the cover letter to make it clear that none of the consulting engagements are related to the work reported in this manuscript.

Authors’ Response: We would like to clarify that none of these commercial entities were involved at all in any way in this study, and these commercial entities did not employee nor pay the salaries of the investigators involved in this study. We have included a sentence about this in the revised cover letter.

Authors’ Response: As described above, we have included updated funding statements and competing interests statements in the revised cover letter.

“This study was primarily funded by a philanthropic gift from Charles W. Swanson to CL. MSD was funded by NIH/NHLBI K12 HL143961 and K23HL172699 (MSD). PYH was supported by NIH/NAID 2K24AI112393. The funders played no role in the study design, data collection and analysis, decision to publish, or preparation of the manuscript. Its contents are solely the responsibility of the authors and do not necessarily represent the official views of the NIH. “

“This study was primarily funded by a philanthropic gift from Charles W. Swanson to CL. MSD was funded by NIH/NHLBI K12 HL143961 and K23HL172699 (MSD). PYH was supported by NIH/NAID 2K24AI112393. The funders played no role in the study design, data collection and analysis, decision to publish, or preparation of the manuscript. Its contents are solely the responsibility of the authors and do not necessarily represent the official views of the NIH.”

Authors’ Response: We have removed these from the manuscript and noted them in the revised cover letter.

Authors’ Response: We have added captions to the supporting information and corrected the in-text citations and file names for the supporting information.

Authors’ Response: Not applicable, reviewers did not suggest citing specific work.

Authors’ Response: We have reviewed the reference list; we have added several references in response to the Editor and Reviewer comments to expand the introduction/background. None of the references we have cited have been retracted.

Additional Editor Comments (if provided):

It is definitely an interesting article which will add value to the current research on COVID-19 studies. However, Please justify the background of the study and how tilt table test adds value.

Authors’ Response: Thank you for this comment. We have added additional background to the introduction.

Reviewers' comments:

Reviewer's Responses to Questions

Comments to the Author

1. Is the manuscript technically sound, and do the data support the conclusions?

Reviewer #1: Partly

Reviewer #2: Yes

2. Has the statistical analysis been performed appropriately and rigorously?

Reviewer #1: Yes

Reviewer #2: Yes

3. Have the authors made all data underlying the findings in their manuscript fully available?

Reviewer #1: Yes

Reviewer #2: Yes

4. Is the manuscript presented in an intelligible fashion and written in standard English?

Reviewer #1: Yes

Reviewer #2: Yes

5. Review Comments to the Author

Reviewer #1: Hello.

This is a good article and the topic chosen is important and special and has been studied in a proper way in this article.

I thank the authors.

I hope we will read more and better quality studies from them in the future.

Good luck.

Author’s Response: We would like to thank Reviewer #1 for their comments regarding the value of our study.

Reviewer #2: The study titled ‘Case-control study of autonomic symptoms in the setting of long COVID with tilt table testing’ by Durstenfeld et al. presents several intriguing findings. To further strengthen the manuscript, I recommend the following revisions:

Authors’ Response: Thank you for your interest in our manuscript and these helpful suggestions.

1. Page 3, Introduction Section: The introduction outlines prior findings and notes inconsistencies in the literature; however, it does not clearly highlight why this represents a significant clinical or scientific problem worth investigating. Beyond mentioning variability in prevalence estimates, the rationale for the study is not compelling. The authors should better articulate the importance of clarifying autonomic dysfunction in Long COVID, to more convincingly justify the need for the present study.

Authors’ Response: Thank you for asking us to highlight the significance of this problem and the justification for our study. We have rewritten the introduction to focus on clinical significance of the possible connection between autonomic symptoms and Long COVID. The new first paragraph of the introduction is as follows (Lines 45-55):

Autonomic disorders including postural orthostatic tachycardia syndrome (POTS), orthostatic hypotension (OH), and orthostatic intolerance without tachycardia or hypotension have been reported following COVID-19 as post-acute sequelae of COVID-19 or symptomatic Long COVID [1, 2]. The role of autonomic dysfunction as a contributor to Long COVID symptomatology is uncertain, but there is overlap between common Long COVID symptoms and common autonomic symptoms. A survey of individuals recruited through Long COVID support groups reported that two thirds of survey participants had a Composite Autonomic Symptom 31 (COMPASS-31) score >20, suggestive of autonomic dysfunction [3]. More severe autonomic symptoms are associated with worse quality of life and higher levels of disability among those with Long COVID [4]. Orthostatic intolerance is included in the diagnostic criteria for myalgic encephalitis/chronic fatigue syndrome (ME/CFS), which overlaps with Long COVID especially in its most disabling forms [5].

2. Page 3, Lines 51-55: When discussing the prior case series, the manuscript briefly notes the influence of beta blocker use and nitroglycerin provocation on HUTT outcomes. However, this limitation is not sufficiently emphasized. The authors should explicitly state that these methodological issues undermine the validity of the reported prevalence of orthostatic intolerance and contribute to inconsistencies across previous studies. This clarification is important to strengthen the rationale for why the present case-control study is warranted.

Authors’ Response: We have expanded the introduction to highlight the methodologic concerns and limitations of the prior studies (new Lines 56-70, with new text bold and underlined):

Head up tilt-table testing (HUTT) is the gold-standard objective test for assessing orthostatic symptoms in the setting of autonomic dysfunction, but existing studies have major limitations. A case-series of 24 individuals with cardiovascular phenotype post-acute sequelae of COVID-19, or Long COVID, referred from a Long COVID clinic, reported that 23/24 had “orthostatic intolerance” during HUTT. In that study, 15 (65%) had normal tests prior to administration of nitroglycerin and 22% were on beta blockers [6]. Beta blockers lower sensitivity and nitroglycerin lowers specificity for clinical symptoms [7]. Importantly, all 15 with nitrog

---

## [Decision Letter · Decision Letter 1]

8 Oct 2025

Case-control study of autonomic symptoms in the setting of long COVID with tilt table testing

PONE-D-25-34093R1

Dear Dr. Durstenfeld,

We’re pleased to inform you that your manuscript has been judged scientifically suitable for publication and will be formally accepted for publication once it meets all outstanding technical requirements.

Kind regards,

Jeyasakthy Saniasiaya, MD, MMed ORLHNS, FEBORLHNS

Academic Editor

PLOS ONE

Additional Editor Comments (optional):

Reviewers' comments:

Reviewer's Responses to Questions

**Comments to the Author**

Reviewer #1: All comments have been addressed

Reviewer #2: All comments have been addressed

2. Is the manuscript technically sound, and do the data support the conclusions?

Reviewer #1: Partly

Reviewer #2: Yes

3. Has the statistical analysis been performed appropriately and rigorously?

Reviewer #1: Yes

Reviewer #2: Yes

4. Have the authors made all data underlying the findings in their manuscript fully available?

Reviewer #1: Yes

Reviewer #2: Yes

5. Is the manuscript presented in an intelligible fashion and written in standard English?

Reviewer #1: Yes

Reviewer #2: Yes

Reviewer #1: Hello

This is a good article and the authors have researched a good topic.

Considering the importance of the coronavirus disease, articles related to this topic are very important.

I hope we will see better and more studies from the authors of this article in the future.

Good luck.

Reviewer #2: (No Response)

**Do you want your identity to be public for this peer review?** For information about this choice, including consent withdrawal, please see our Privacy Policy

Reviewer #1: No

Reviewer #2: No

---

## [Editor Report · Acceptance letter]

PONE-D-25-34093R1

PLOS ONE

Dear Dr. Durstenfeld,

I'm pleased to inform you that your manuscript has been deemed suitable for publication in PLOS ONE. Congratulations! Your manuscript is now being handed over to our production team.

Kind regards,

on behalf of

Dr. Jeyasakthy Saniasiaya

Academic Editor

PLOS ONE